# The Incidence of Chronic Kidney Disease Three Years after Non-Severe Acute Kidney Injury in Critically Ill Patients: A Single-Center Cohort Study

**DOI:** 10.3390/jcm8122215

**Published:** 2019-12-14

**Authors:** Sébastien Rubin, Arthur Orieux, Benjamin Clouzeau, Claire Rigothier, Christian Combe, Didier Gruson, Alexandre Boyer

**Affiliations:** 1Service de Néphrologie, Transplantation, Dialyse, Aphérèses, Hôpital Pellegrin, Centre Hospitalier Universitaire de Bordeaux, 33076 Bordeaux CEDEX, France; 2Service de Médecine Intensive Réanimation, Hôpital Pellegrin, Centre Hospitalier Universitaire de Bordeaux, 33076 Bordeaux Cedex, France

**Keywords:** acute kidney injury, critically ill patients, renal recovery, chronic kidney disease, end stage renal disease

## Abstract

The risk of chronic kidney disease (CKD) following severe acute kidney injury (AKI) in critically ill patients is well documented, but not after less severe AKI. The main objective of this study was to evaluate the long-term incidence of CKD after non-severe AKI in critically ill patients. This prospective single-center observational three-years follow-up study was conducted in the medical intensive care unit in Bordeaux’s hospital (France). From 2013 to 2015, all patients with severe (kidney disease improving global outcomes (KDIGO) stage 3) and non-severe AKI (KDIGO stages 1, 2) were enrolled. Patients with prior eGFR < 90 mL/min/1.73 m^2^ were excluded. Primary outcome was the three-year incidence of CKD stages 3 to 5 in the non-severe AKI group. We enrolled 232 patients. Non-severe AKI was observed in 112 and severe AKI in 120. In the non-severe AKI group, 71 (63%) were male, age was 62 ± 16 years. The reason for admission was sepsis for 56/112 (50%). Sixty-two (55%) patients died and nine (8%) were lost to follow-up. At the end of the follow-up the incidence of CKD was 22% (9/41); Confidence Interval (CI) _95%_ (9.3–33.60)% in the non-severe AKI group, tending to be significantly lower than in the severe AKI group (44% (14/30); CI _95%_ (28.8–64.5)%; *p* = 0.052). The development of CKD three years after non-severe AKI, despite it being lower than after severe AKI, appears to be a frequent event highlighting the need for prolonged follow-up.

## 1. Introduction

Acute kidney injury (AKI) is very common in Intensive Care Unit (ICU) patients since it is estimated to develop in up to 50% of them [1]. The short-term implications have been studied extensively and include an increase in mortality [2], hospitalization length, rehospitalizations [3], and impaired quality of life [4]. The long-term implications in critically-ill patients are still not well known, except for severe AKI (defined most of the time by AKI-requiring renal replacement therapy (RRT)). In that category of patients, Schiffl et al. (2008) found that five years after RRT, 14% of them developed chronic kidney disease (CKD) [5]. Gammelager et al. (2013) demonstrated that among surviving patients requiring RRT, the incidence of end-stage renal disease (ESRD) at five years was 4% [6]. 

To date, non-severe AKI outcomes were predominantly studied in non-critically ill patients, using large administrative data sets [7], e.g., veterans’ health administration data [8]. These analyses are known to have low sensibility [9] and to underestimate the less severe form of AKI [10]. Another big cohort including non-severe AKI in non-critically ill patients is the 5-year prospective case-control ‘AKI Risk in Derby’ study. Patients were divided into two groups according to the kidney disease improving global outcomes (KDIGO) classification (severe AKI corresponding to KDIGO 3 and non-severe AKI corresponding to KDIGO 1 or 2) [11]. Preliminary results after a 12-month follow-up showed that patients with severe AKI had worse kidney function than patients with non-severe AKI, but kidney function declined in both groups compared to control patients [12]. These studies did not enroll many critically ill patients who typically present with a different spectrum of AKI etiologies [13]. These patients are often exposed to multiple nephrotoxic agents (antibiotics, iodine contrast products, etc.), severe hemodynamic variations, and inflammatory state (“sepsis”), each being able to worsen the long-term renal prognosis.

To date, it is difficult to determine the real risk of developing CKD in critically ill patients with non-severe AKI. The incidence of CKD many years after a non-severe AKI in ICU could be thus underestimated. The main objective of this study was to evaluate the long-term incidence of CKD after non-severe AKI in critically ill patients. This would be the first study to address this issue. Secondary objectives were to compare CKD incidence after non-severe vs. severe AKI episodes, to evaluate risk factors for developing CKD, and to identify the proportion among CKD patients followed by a nephrologist.

## 2. Materials and Methods

### 2.1. Study Design

This prospective three-year follow-up observational study was carried out in Bordeaux, France, from September 2013 to May 2015. Our center participated in the artificial kidney initiation in kidney injury (AKIKI) study [14], during which all patients with AKI from stage 1 of the KDIGO classification were prospectively and carefully screened. Data were collected during the period of hospitalization. After discharge, the follow-up was carried out three years after enrollment. A direct contact with the general practitioner (GP) and/or the patient was achieved. According to French law, the database was declared to the French data protection authority (declaration number 2168624). The study obtained the approval of the ethics commission of the French society of intensive care medicine and was assigned as CE SRLF 18-20.

### 2.2. Participants

Patients were enrolled if they were 18 years of age or older, received invasive mechanical ventilation, catecholamine infusion, or both, and developed AKI assessed by an increase of serum creatinine (SCr) of >26.5 µmol/L within 48 h or an increase of >1.5 times the baseline value, according to KDIGO guidelines [11]. Patients with an estimated glomerular filtration rate (eGFR) < 90 mL/min/1.73 m^2^ prior to ICU admission, using the chronic kidney disease epidemiology collaboration (CKD-EPI) formula, were excluded. Serum creatinine assays were standardized (IDMS calibration). Enrollment and collection of hospitalization data were recorded prospectively. Follow-up outcomes were collected prospectively three years after enrollment for each patient, in order to limit memorization bias or loss of data, using medical records and phone calls with GPs and patients.

### 2.3. Acute Kidney Injury Classification

The KDIGO staging of AKI was used to define non-severe AKI (AKI stages 1 and 2) and severe AKI (AKI stage 3) [11]. Only SCr was considered because of inconsistent urine output data. Baseline SCr were SCr at admission in the case of normal renal function or SCr less than 1 year in the case of abnormal SCr at admission. All baseline SCr were obtained by previous blood tests.

### 2.4. Exposure Variables

Information relative to smoker status, past medical history of hypertension, diabetes, chronic heart failure, ischemic heart disease (IHD), stroke, peripheral arterial disease (PAD), prior CKD, reason of admission, simplified acute physiology score II (SAPS II), length of hospitalization, catecholamine, aminoglycoside, contrast agent use, or death were collected using prospectively recorded data and patient questioning if applicable.

Because some patients were treated both with continuous veno-venous hemodialysis (CVVHD) and intermittent hemodialysis (IHD), we only recorded the first RRT modality that was used during the ICU stay. 

### 2.5. Long-Term Incidence of CKD at Three Years

We defined CKD as eGFR <60 mL/min/1.73 m^2^, using the chronic kidney disease epidemiology collaboration (CKD-EPI) corresponding to CKD stage 3 or more according to the KDIGO classification. Creatinine level were collected by family calls or GP calls. In the case of abnormal creatinine level, absence of AKI was checked using an anteriority blood test. 

### 2.6. Other Outcomes

Recovery from AKI was determined at ICU discharge. It was defined as a return of SCr to <26.5 µmol/L (<0.3 mg/dL) above baseline for alive and non-dependent RRT patients. These data were collected using hospital records or using data from blood tests performed outside the hospital, with prior consent of the patient. We called CKD patients to ask if they were followed by a nephrologist. The survival state and date of death if applicable were collected using hospital records, GP, and family phone calls.

### 2.7. Statistical Analysis

Statistical analysis was carried out using JMP^®^ Version 14, SAS Institute Inc., Cary, NC, USA, 1989–2007. Descriptive statistics included mean ± standard deviation (SD) or median (Quartile 1- Quartile 3). Quantitative variables were compared using a *t*-test, and qualitative variables using Chi^2^ Pearson test. The multivariate analysis was carried out using logistic regression. To choose independent variables included in the model, we allowed one independent variable for every 20 patients analyzed. Interactions between independent variables were checked using the Pearson correlation test for quantitative variables and the Chi^2^ test for ordinal or binomial variables using Yates’ correction if the sample size was <10. Quantitative variables were stratified into a range when a constant magnitude of association was not consistent. Renal survival was studied using the Kaplan–Meier curve. Survival curves were compared using a log-rank test. A value of *p* < 0.05 was considered statistically significant (double-sided).

## 3. Results

### 3.1. Participants

From 2013–2015, 304 patients with AKI were admitted to the ICU (Figure 1). Among them, 72 had prior CKD and were excluded and 232 patients were enrolled. No patients had missing baseline creatinine value. Non-severe AKI was present in 112 (AKI stage 1, 62; stage 2, 50) and severe AKI in 120 (AKI stage 3). In the non-severe AKI group, 71/112 (63%) were male with a mean age of 62 ± 16 years. In 56/112 (50%), the reason for admission was sepsis, 89/112 (79%) required catecholamines, and 92/112 (79%) were intubated. The simplified acute physiology score II (SAPS II) was 59 ± 17. The duration of hospitalization was 9 ± 10 days. All descriptive characteristics and the comparison between non-severe AKI vs. severe AKI are presented in Table 1. In severe AKI, patients had a higher SAPS II (65 ± 20 vs. 59 ± 17; *p* = 0.01). The ICU mortality rate was also higher in the severe patients group (57/120 (48%) vs. 34/112 (30%); *p* = 0.01). Renal replacement therapy was performed in 73/120 (61%) of patients with severe AKI, and CVVHD was used in 57/73 (78%) of these patients. One hundred and seven patients died in hospital (40 in the non-severe group and 67 in the severe group). Cause of death was multiple organ failure (35 patients), withholding or withdrawing of life-prolonging therapy (16), neurologic disorder (15), septic shock (11), acute respiratory distress syndrome (10), hemorrhagic shock (3), cardiac arrest (4), or others (13). Among the 78/112 (69%) patients with non-severe AKI who survived at ICU discharge, renal recovery was observed in 68/78 (87%) patients compared to 22/63 (35%) patients with severe AKI (*p* < 0.001). 

Descriptive characteristics of all patients enrolled (patients enrolled) and comparative descriptive characteristics between non-severe AKI and severe AKI. Statistical analysis was carried out to compare these two subgroups. 

### 3.2. Follow-Up

A flow diagram is presented in Figure 1. In non-severe AKI, 34/112 (30%) died in the ICU and 28/112 (25%) during the follow-up. We lost 9/112 (8%) patients to follow-up. In this group, 41 patients completed the study.

### 3.3. Primary Outcome: Long-Term Incidence of CKD at Three Years

In non-severe AKI, the incidence of CKD during a three years follow-up amongst patients who survived was 9/41 (22% CI _95%_ (9.3–33.6)). It tended to be lower than in in the severe AKI group (14/30 (44% CI _95%_ (28.8–64.5)) *p* = 0.052). Among the 23 patients who developed CKD, whatever the group, 8 had recovered from AKI (6 in the non-severe group) and 9 had eGFR < 60 mL/min/1.73 m^2^ (4 in the non-severe group) at ICU discharge. CKD stages at three years are summarized in Table 2. 

### 3.4. Secondary Outcomes

#### 3.4.1. Risk Factors for CKD at Three Years

In the univariate analysis, hypertension (Odd Ratio (OR) = 3.5 (1.2–10.5)), diabetes (OR = 3.6 (1.2–10.3)), SCr (OR = 1.007 (1.002–1.011)), and severe AKI (OR = 3 (1.1–8.5)) were significantly associated with CKD at three years.

In the multivariate analysis, hypertension and diabetes presented interactions (Chi^2^
*p* = 0.004), as well as Scr and severe AKI. Diabetes and severe AKI were the variables maintained in the analysis. In this model, only diabetes (OR = 3.3 (1.3–8.3)) was significantly associated with CKD at three years. Conversely, the severity of AKI was not associated with CKD (severe vs. non-severe) (OR = 1.96 (0.8–5)) (Table 3).

#### 3.4.2. Patients Survival

Patients survival was assessed in the set of 232 included patients. At three years, survival was 38% in our series. The three years survival was 43% in the non-severe AKI group and 32% in the severe AKI group with statistical difference (*p* = 0.02) (Figure 2). 

#### 3.4.3. Renal Specialist Following

Eleven out of twenty-three (48%) patients who developed CKD were followed by a nephrologist.

## 4. Discussion

This study is the first in the literature to estimate the incidence of CKD three years after non-severe AKI in critically ill patients. At three years, an eGFR of <60 mL/min1.73 m^2^ (defining CKD) was present in 22% CI _95%_ (9.3–33.6) in the non-severe AKI group, half of whom were not followed by a nephrologist.

Our study is original. The incidence of CKD only three years after non-severe AKI is high (22%). No study has focused on stage 1 and 2 AKI in critically ill patients while only a few studies have studied stage 1 and 2 AKI in non-critically ill patients. In the recent analysis of U.S. veterans’ health administration data, incidence of CKD (eGFR < 60 mL/min/1.73 m^2^) at 1 year was 31% in AKI stage 1-patients and 27% in AKI stage 2-patients [15]. However, patients had more risk factor to develop CKD than ours; they were male (95%), older than ours, and basal eGFR was 84 mL/min/1.73 m^2^. We excluded patients with an eGFR of <90 mL/min/1.73 m^2^ prior to AKI Many similar studies excluded patients with prior eGFR < 60 mL/min/1.73 m^2^. Indeed, patients with an eGFR of 60–90 mL/min/1.73 m^2^ do not have normal renal function and it has been very well demonstrated that even a slight degree of chronic renal failure promotes future alteration of eGFR [16]. By excluding patients with DFG < 90 mL/min/1.73 m^2^ at inclusion, we ensure a decrease of 30 mL/min in three years, which is clinically very significant. 

Nevertheless, many factors may influence CKD development three years after an AKI in ICU and we cannot conclude whether non-severe AKI itself was independently implicated in the high CKD incidence at three years. First, AKI itself: in vitro studies have highlighted different mechanisms linking AKI and CKD, such as persistent interstitial inflammation or tubular’s vascular damages [16]. Secondly, individual factor can lead to the development of CKD at long-term. For example, diabetes is associated with a decrease in kidney function in many studies. In the multivariate analysis, diabetes was a risk factor for a decline in kidney function in AKI patients without prior CKD. These results were already demonstrated in non-critically ill patients. However, many other clinical conditions are associated with the risk of developing CKD such as age, sex, diabetes, hypertension, albuminuria, initial eGFR, high triglyceride levels, and low HDL cholesterol levels [17,18]. In a future study, the Kidneyfailurerisk.com Canadian score, which estimates the risk to develop CKD at 2 or 5 years and is well validated in a variety of populations [19], could be used to compare the expected incidence assessed by this score with the observed increased incidence of CKD.

One third of surviving patients had apparent complete renal recovery at ICU discharge but later developed CKD. However, for the other patients, we could not determine if they recovered later or if they kept low eGFR over the three years. These findings are in accordance with studies performed in non-critically ill patients, which found an increased risk of CKD following non-dialysis-dependent AKI, even after biological renal recovery [20,21]. Absence of renal recovery at ICU discharge remains common in our study (about a third of the cases). Kellum et al. studied 17,000 ICU patients with stage 2–3 AKI and showed that early relapse of AKI occurred in 37% of cases. Late sustained reversal (after 7 days and sustained through hospital discharge) and relapse were two risk factors for a decreased age-adjusted one-year renal survival [22].

One of the strength of our study comes from the recording of anterior SCr rather than MDRD estimated SCr. The main limitation of our study is its single-center characteristic with a consecutive low number of surviving patients at three years, which favors a risk of type 2 error. We have screened more than 300 patients in three years. The enrollment was exhaustive because it was integrated in a clinical trial in which the medical team was very involved. Only 22 patients were lost to follow-up (<10%), a satisfactory proportion for this type of study. Despite this, because of CKD occurring before AKI accounted for many exclusions but also because of many deaths, few patients (23%) could be analyzed at the three-year follow-up. This lack of power probably explains the absence of significant association between AKI severity and CKD. This association was already suggested by many studies showing that severe AKI remains the main prognostic factor of CKD after a long follow-up period. We could not determine whether CKD developed before death in the patients who died. However, 109/139 patients (78%) died before day 90, which is the time limit after which CKD can be defined. Our hypothesis is that less severe patients with a longer outcome need a specific follow-up to detect and to prevent CKD. This pragmatic view led us to identify incidence of CKD only in patients who survived at three years.

It is clear that AKI survivors require a long-term follow-up [23]. First, they are at high risk of developing CKD even in cases of non-severe AKI, including if their kidney function has recovered at ICU discharge. Second, this risk is probably underestimated both by the patient and the general practitioner because less than 50% of patients with CKD were followed by a nephrologist in our study. Third, it was already shown that the risk of mortality and cardiovascular complications is very high in patients with an eGFR of <60 mL/min/1.73 m^2^ and the long-term consequences concern other organs and persist despite renal recovery [24,25].

The risk of developing CKD at three years after non-severe AKI, despite it being lower than after a severe AKI, remains high. These findings have to be confirmed by larger studies. A long-term follow up is required and all physicians involved in the patient’s follow-up, including intensivists, should pay attention to that phenomenon.

## Figures and Tables

**Figure 1 jcm-08-02215-f001:**
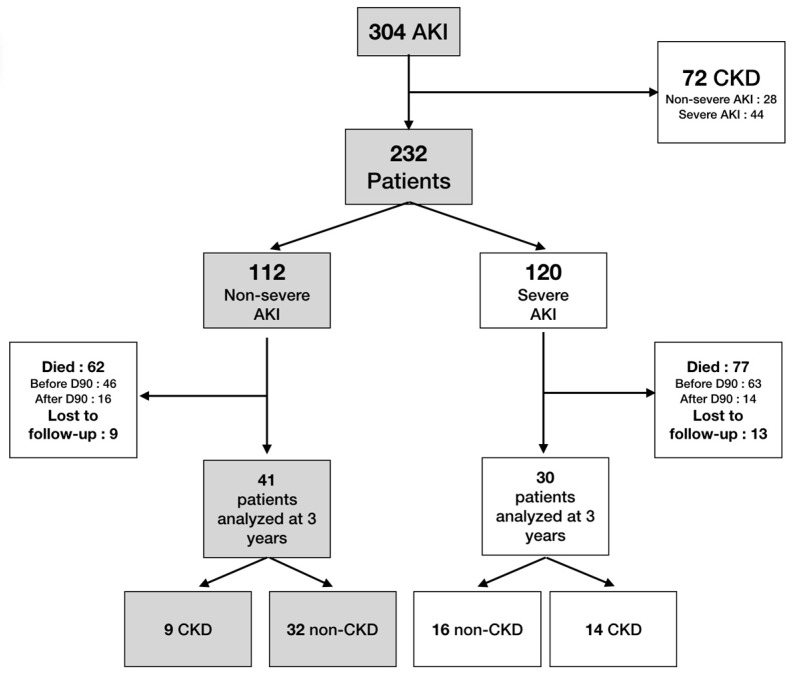
Flow chart. CKD: chronic kidney disease; AKI: acute kidney injury.

**Figure 2 jcm-08-02215-f002:**
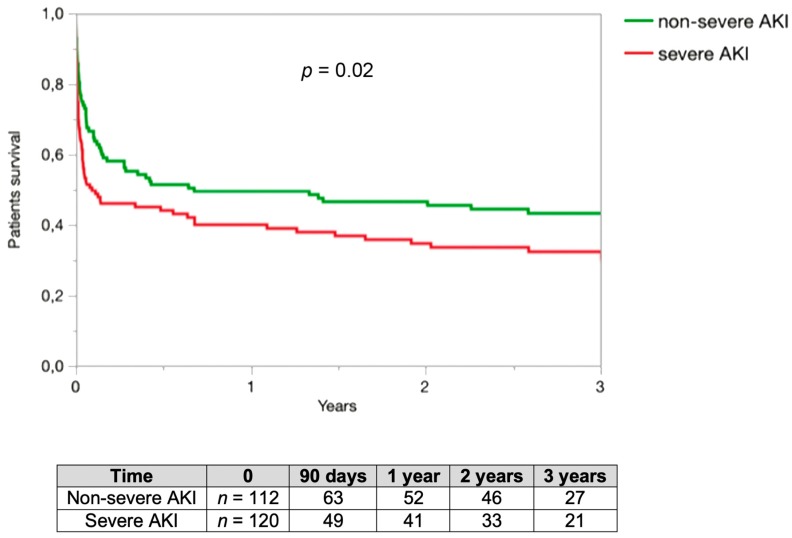
Patients’ survival rate. Renal survival was assessed in 232 patients. The three years renal survival was 43% in the non-severe AKI group and 32% in the severe AKI group with statistical difference. Comparison of renal survival rate using log-rank test.

**Table 1 jcm-08-02215-t001:** Descriptive characteristics of patients enrolled in the cohort and comparative descriptive characteristics of patients with non-severe AKI and severe AKI.

Characteristics of Patients	Patients Enrolled *n* = 232 (%)	Non-Severe AKI *n* = 112 (%)	Severe AKI *n* = 120 (%)	*p* Value
Males	142 (63)	71 (63)	71 (59)	0.5
Age	62 ± 16	62 ± 16	62 ± 16	0.8
Smoker	97 (42)	50 (45)	47 (39)	0.4
Hypertension	115 (50)	62 (55)	53 (44)	0.1
Diabetes	55 (24)	31 (28)	24 (20)	0.2
Heart failure	42 (18)	20 (18)	22 (18)	0.9
Stroke	21 (9)	12 (11)	9 (8)	0.4
PAD	17 (7)	6 (5)	11 (9)	0.3
IHD	33 (14)	19 (17)	14 (11)	0.2
Basal SCr	78 ± 18	78 ± 19	77 ± 17	0.6
Sepsis	118 (51)	56 (50)	62 (52)	0.8
Contrast agent	55 (24)	26 (23)	29 (24)	0.9
Aminosid use	84 (36)	43 (38)	42 (35)	0.6
NIV or HFNC	192 (83)	6 (5)	5 (4)	0.7
Orotracheal intubation	192 (83)	92 (82)	100 (83)	0.8
Catecholamine use	191 (82)	89 (79)	102 (85)	0.3
SAPS II	62 ± 19	59 ± 17	65 ± 20	0.01
Maximal SCr (µmol/L)	266 ± 181	153 ± 56	371 ± 195	<0.001
AKI stage:				
1	62 (27)	62 (55)	0 (0)	
2	50 (21)	50 (45)	0 (0)	
3	120 (52)		120 (100)	
RRT		0	73 (61)	
CVVHD		0	57 (48)	
IHH		0	16 (13)	
Renal recovery	90/141 (64)	68/78 (87)	22/63 (35)	<0.001
ICU length of stay (days)	9 ± 10	9 ± 10	9 ± 11	0.9
Intra-ICU deaths	91 (39)	34 (30)	57 (48)	0.01
Hospital length of stay (days)	36 ± 100	37 ± 109	34 ± 91	0.8

PAD: Peripheral arterial disease; IHD: Ischemic heart disease; HFNC: High-flow nasal cannula; NIV: Non-invasive ventilation; AKI: Acute kidney injury; SAPS II: Simplified acute physiology score II; SCr: Serum creatinine; ICU: Intensive care unit; RRT: renal replacement therapy; CVVHD: Continuous venovenous hemodialysis; IH: Intermittent hemodialysis.

**Table 2 jcm-08-02215-t002:** Chronic kidney disease stage at three years follow-up (eGFR (mL/min/1.73 m^2^)).

CKD Stages At 3 Years	Non-Severe AKI at Inclusion *n* = 41 (%)	Severe AKI at Inclusion *n* = 30 (%)	Total *n* = 71 (%)
CKD3 (60 < eGFR < 30)	7 (17)	10 (33)	17 (24)
CKD4 (30 < eGFR < 15)	2 (5)	1 (3)	3 (4)
CKD5 (eGFR < 15))	0	3 (10)	3 (4)

AKI: acute kidney injury; CKD: chronic kidney disease.

**Table 3 jcm-08-02215-t003:** Risk factors for developing CKD at three years.

	Univariate Analysis		Multivariate Analysis	
	Odds Ratio eGFR < 60 (mL/min/1.73 m^2^)	Confidence Interval 5%	Odds Ratio eGFR < 60 (mL/min/1.73 m^2^)	Confidence Interval 5%
Male	0.5	(0.2–1.4)		
Age	1.1	(0.99–1.2)		
Smoker	1.8	(0.6–4.8)		
Hypertension	3.5	(1.2–10.5)		
Diabetes	3.6	(1.2–10.3)	3.3	(1.3–8.3)
Heart failure	2.3	(0.6–9.1)		
Stroke	0.3	(0.04–2.7)		
PAD	2.1	(0.3–16.3)		
IHD	1.8	(0.4–7.3)		
Sepsis	1.7	(0.6–4.6)		
Contrast agent	1.3	(0.4–3.8)		
Aminosid use	2.1	(0.8–5.8)		
Orotracheal intubation	0.7	(0.2–2.1)		
Vasopressor	1.6	(0.4–6.5)		
SAPS II	1.5	(0.2–12.9)		
Length of hospitalization in ICU (days)	0.97	(0.94–1.04)		
Hospital length of stay (days)	0.99	(0.98–1.01)		
Maximum SCr	1.007	(1.002–1.01)		
Non-severe AKI	1		1	
Severe AKI	3	(1.1–8.5)	1.96	(0.8–5)
AKI stage 1	0.2	(0.05–0.4)		
AKI stage 2	0.5	(0.2–1.5)		
AKI stage 3	1			
RRT	2.7	(0.9–8.2)		
CVVHD	0.8	(0.2–3.1)		
Readmission at hospital during follow-up	1.5	(0.5–4.3)		

Multivariate analysis was proceeded using logistic regression. PAD: Peripheral arterial disease; IHD: Ischemic heart disease; HFNC: High-flow nasal cannula; NIV: Non-invasive ventilation; AKI: Acute kidney injury; SAPS II: Simplified Acute physiology score; SCr: Serum creatinine; CVVHD: Continuous veno-venous hemodialysis; RRT: Renal replacement therapy; ICU: Intensive care unit.

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
