# Peer review of "The Incidence of Chronic Kidney Disease Three Years after Non-Severe Acute Kidney Injury in Critically Ill Patients: A Single-Center Cohort Study"

_jcm, 2019, doi:10.3390/jcm8122215_

Round 1

Reviewer 1 Report

Please see the attached file. Thanks!

Author Response

Reply to the review report

Reviewer 1

The reviewer 1 mainly commented on the limited number. We agree that the major limitation is the low case number. Our aim was to provide an early warning signal concerning the risk for patients suffering from non-severe AKI to develop CKD. Therefore, this warning should immediately encourage intensivists to carry out systematic follow-up of these patients despite an apparent recovery. Indeed, these findings have to be confirmed by larger studies. We have added this sentence in the conclusion: “These findings have to be confirmed by larger studies.”

Secondly, he said that “Many factors may influence the results. Episodes of kidney injury, chronic exposure to nephrotoxic drugs and newly developed comorbidity were difficult to clarify”. He is correct, and we thank him for this observation. In line with this comment and another reviewer comment, we have modified the introduction and added the following paragraph in the discussion. These modifications added 140 word in the manuscript.

Introduction:

“In addition, these patients are often exposed to multiple nephrotoxic (antibiotics, iodine contrast products, etc.), severe hemodynamic variations, and inflammatory state ("sepsis") that can worsen the long-term renal prognosis.”

Discussion:

“Nevertheless, many factors may influence CKD development 3 years after an AKI in ICU and we cannot conclude whether non-severe AKI itself was independently implicated in the high CKD incidence at 3 years. First, AKI itself: in vitro studies have highlighted different mechanisms linking AKI and CKD such as persistent interstitial inflammation or tubular’s vascular damages. Secondly, individual factor can lead to the development of CKD at long-term. For example, diabetes is associated with a decrease in kidney function in many studies. In the multivariate analysis, diabetes was a risk factor for a decline in kidney function in AKI patients without prior CKD. These results were already demonstrated in non-critically ill patient. However, many other clinical conditions are associated with the risk of developing CKD such as age, sex, diabetes, hypertension, albuminuria, initial eGFR, high triglyceride levels and low HDL cholesterol levels[17][18]. In a future study, the Kidneyfailurerisk.com Canadian score which estimates the risk to develop CKD at 2 or 5 years and is well validated in a variety of populations [19] could be used to compare the expected incidence assessed by this score with the observed increased incidence of CKD.”

Moreover, he noted that “compared CKD incidence according to initial kidney injury and grouped as non-severe and severe injury group in this study. However, part of survived patients had their GFR <60 ml/min since then. We cannot find these details in the study. Moreover, for those who recovered from non-severe or severe AKI, the difference of their renal progression or incidence of CKD may be more important”.

We fully agree with this comment. Indeed, a group of patients with AKI kept eGFR<60ml/min/1.73m2 until the 3 years evaluation, whereas another group first improved renal function from ICU discharge then deteriorated it later. We have performed a new analysis and we found that among the 23 patients who had an eGFR<60ml/min/1.73m2 at 3 years (9 in the non-severe vs 14 in the severe AKI groups): 9 (39%) had eGFR<60ml/min/1.73m2 at ICU discharge which may represent the first group of patient. This proportion was 4/9 (44%) in the non-severe AKI group and 5/14 (35%) in the severe AKI group.

Consequently, we added this information in the “result section” line  226

“and 9 had eGFR<60ml/min/1.73m2 (4 in the non-severe group) at ICU discharge”

And we also added a comment in the discussion part (line 307):

“However for the other patients, we could not determine if they recovered later or if they kept low eGFR along the 3 years”.

He asked “In Table 3, how to interpret the result about the impact of non-severe AKI on CKD risk?”.

Despite statistically significant in the univariate analysis, severe AKI only tends to increase the risk of developing CKD compared to non-severe AKI. The low case number is responsible for a lack of power that could partly explain this result (confusion bias being the other potential explanation). Beyond the fact that statistical significance was not reached, it is not surprising  that the severity of the initial injury to the kidney, i.e. the severity of AKI, was relied to CKD development. This is consistent with the physiological link between AKI and CKD. According to the Reviewer point, we propose to modify the initial comment wrote in the first version of the of the discussion, i.e.:

“For example, AKI severity was not significantly associated with an eGFR of <60 ml/min/1.73m2 at 3 years in the multivariate analysis. This result is not in accordance with many studies showing that severe AKI remains the main prognostic factor of CKD after a long follow-up period”

which becomes now (line 323):

“This lack of power probably explains the absence of significant association between AKI severity and CKD. This association was already suggested by many studies showing that severe AKI remains the main prognostic factor of CKD after a long follow-up period”.

Last, he asked “In table 3, Why the RRT and CVVHD were listed separately?”.

We made this distinction because some authors have suggested that CVVHD can be associated with an improved renal outcome 1,2,3. This was not the case in our study.

Vinsonneau C. et al, Lancet, 2006 : 10.1016/S0140-6736(06)69111-3 Mehta R, McDonald B, Gabbai FB, Pahl M, Pascual MT, Farkas A, et al. A randomized clinical trial of continuous versus intermittent dialysis for acute renal failure. Kidney Int 2001;60:1154–63. Uehlinger DE, Jakob SM, Ferrari P, Eichelberger M, Huynh-Do U, Marti HP, et al. Comparison of continuous and intermittent renal replacement therapy for acute renal failure. Nephrol Dial Transplant 2005;20:1630–7.

Reviewer 2 Report

I read with interest the paper entitled “The incidence of chronic kidney disease three years after non-severe acute kidney injury in critically ill patients: a single center cohort study”. Unfortunately in the present form I do not think it could be published.  The points that should be clarified are:

The title focus is on non-severe acute kidney injury (AKI), however authors enrolled also patients with severe AKI. The sentences in the introduction section included in lines 50-55 are not clear, I would rather focus on risk of development of chronic kidney disease (CKD). In my opinion study design is retrospective, authors appeared to collect different information at the beginning of the study (mainly cardiovascular comorbidity), but cause of death during follow-up is lacking. If serum creatinine was assessed by standardized calibration, what’s the meaning of writing “No SCr values were estimated using formula”? All abbreviation in the text, tables and figures should be explained. Primary, secondary and modelling should be clearly stated, and numbers of patients in the non-severe and severe AKI groups should be clearly checked in the entire paper. Patients analysed at three years change in figure 1 and table 2. In table 2 the counting of non-severe AKI is 29 and the counting of severe AKI is 25, totally 54, quite different from numbers written by authors. Reading the text I could understand that 10 patients in the non-severe group and 41 in the severe one had no renal recovery, moreover adding 34+46+16+9 in the non-severe group we obtain 105, but 105+41 is non 112. The same happens in the severe group. I strongly disagree with calculation of chronic kidney disease in the two groups: 112 had non-severe AKI, of whom 9 had chronic kidney disease at three years (8%); 120 had severe AKI, of whom 14 had chronic kidney disease at three years (11.6%). These results sound different from those reported by authors. In my opinion Figure 2 is related to patients’ survival, in fact the numbers reported at 90 days are the same of the deaths at 90 days, therefore the figure should be renamed from renal survival in patients’ survival. The paragraph named “Renal specialist following” is useless. References should follow the journal style. Discussion should be rewritten taken into consideration risk factors for development of chronic kidney disease as suggested by different papers, eventually relating the items to critical ill patients.

Suggested references

O’Seaghdha CM, Lyass A, Massaro JM, Meigs JB, Coresh J, D’Agostino RB, Astor BC, Fox CS. A Risk Score for Chronic Kidney Disease in the General Population. Am J Med 2012; 125(3):270–277. doi: 10.1016/j.amjmed.2011.09.009

Carrillo-Larco, R.M., Miranda, J.J., Gilman, R.H. et al. Risk score for first-screening of prevalent undiagnosed chronic kidney disease in Peru: the CRONICAS-CKD risk score. BMC Nephrol 2017;18, 343. doi:10.1186/s12882-017-0758-4

McMahon GM, Preis SR, Hwang S-J, Fox CS. Mid-Adulthood Risk Factor Profiles for CKD. JASN 2014;25(11):2633-2641; DOI: https://doi.org/10.1681/ASN.2013070750

https://kidneyfailurerisk.com/

Tangri N, Stevens LA, Griffith J, et al. A predictive model for progression of chronic kidney disease to kidney failure. JAMA 2011;305(15). DOI:10.001/jama.2011.451

Tangri N, Grams ME, Levey AS et al. Multinational Assessment of Accuracy of Equations for Predicting Risk of Kidney Failure: A Meta-analysis. JAMA 2016;315(2):1-11. doi:10.1001/jama.2015.18202

Echouffo-Tcheugui JB, Andre P. Kengne AP. Risk Models to Predict Chronic Kidney Disease and Its Progression: A Systematic Review. PLoS Med  2012:9(11): e1001344. doi:10.1371/journal.pmed.1001344

Collins G, Altman. D Predicting the risk of chronic kidney disease in the UK: an evaluation of QKidney® scores using a primary care database. British Journal of General Practice 2012;62(597): e243-e250. doi: https://doi.org/10.3399/bjgp12X636065

Wan-Chuan T, Hon-Yen W, Yu-Sen P, Mei-Ju K, Ming-Shiou W, Kuan-Yu H, Kwan-Dun W, Tzong-Shinn  C,  Kuo-Liong C. Risk Factors for Development and Progression of Chronic Kidney Disease A Systematic Review and Exploratory Meta-Analysis. Medicine 2016;95(11):e3013. doi: 10.1097/MD.0000000000003013

Author Response

Reply to the Review Report
Reviewer 2
Reviewer 2 provided us several main comments and we responded to each of them
ď‚· “The title focus is on non-severe acute kidney injury (AKI), however authors enrolled also patients with severe AKI”
The main objective of our study was the incidence of CKD 3 years after non-severe AKI. However, since the association between severe AKI and CKD is known, we intended to compare severe vs. non-severe AKI and to seek for a potential dose effect. This secondary objective led to the enrolment of patients with severe AKI.
ď‚· “The sentences in the introduction section included in lines 50-55 are not clear. I would rather focus on risk of development of chronic kidney disease (CKD)”
In line with another reviewer comment, we have rewritten this section.
The initial version, i.e.:
“These studies did not enroll many critically ill patients who typically present with a different spectrum of AKI etiologies (more sepsis and hemodynamic failure, but fewer iatrogenic causes) [13]. These distinctions may lead to different histological impairments leading to subsequent long term prognostic modifications”
becomes now (line 77) :
“These studies did not enroll many critically ill patients who typically present with a different spectrum of AKI etiologies [13]. These patients are often exposed to multiple nephrotoxic agents (antibiotics, iodine contrast products, etc.), severe hemodynamic variations and inflammatory state ("sepsis") each being able to worsen the long-term renal prognosis.”
ď‚· “In my opinion study design is retrospective, authors appeared to collect different information at the beginning of the study (mainly cardiovascular comorbidity), but cause of death during follow-up is lacking”
The Reviewer is correct in pointing out that we did not specify the causes of death. However, we can correct this point since we prospectively assessed all the causes of death if they occurred in the hospital before discharge. We could not assess the cause of death after discharge, because the ethical review board only authorized us to obtain the patient’s renal status at 3 years. We have added the cause of Hospital death in the results section: (line 188)
“One hundred and seven patients died in ICU (40 in the non-severe group and 67 in the severe group) Cause of death was multiple organ failure (35 patients), withholding or withdrawing of life-prolonging therapy (16), neurologic disorder (15), septic shock (11), acute respiratory distress syndrome (10), hemorrhagic shock (3), cardiac arrest (4), or others (13).”
ď‚· “If serum creatinine was assessed by standardized calibration, what’s the meaning of writing “No SCr values were estimated using formula”? “
In multiple studies in intensive care patients, basal creatinine is estimated using MDRD formula with a hypothesized eGFR =75 ml/min/1.73m2. One of the strength of our study is to have collected basal creatinine (<1 year) for all enrolled patient (using previous blood test, most often recorded in the Hospital Medical record). Due to the absence of missing data, we did not need to use estimated creatinine using a formula. Following this suggestion, we have deleted this sentence from our manuscript and added “All baseline SCr were obtained by previous blood tests.” (line 115)
ď‚· “All abbreviation in the text, tables and figures should be explained”
To take this suggestion into account, we have added clarifications to the abbreviations that were previously unexplained in figure 1 ant table 1.
ď‚· “Primary, secondary and modelling should be clearly stated, and numbers of patients in the non-severe and severe AKI groups should be clearly checked in the entire paper.“
We understand the misunderstanding probably because the way we present figure 1 and Table 2 can be confusing. However, there was no mistake in the number of patients after we checked them. We hereby respond to each specific comment of the Reviewer concerning the numbers of patients, including the new Figure 1 and Table 2.
a. “Patients analysed at three years change in figure 1 and table 2. In table 2 the counting of non-severe AKI is 29 and the counting of severe AKI is 25, totally 54, quite different from numbers written by authors”
Table 2 shows renal outcome at 3 years graded by CKD stage. Patients with eGFR>90 ml/min/1.73m2 at 3 years were not mentioned in table 2. For example, for non-severe AKI, 41 patients were analyzed at 3 years (figure 1 and table 2 (row 1; column 2; same value). Among them, 29 had eGFR<90ml/min and 9 had eGFR<60ml/min (primary criteria in our study) at 3 year. To avoid similar misunderstanding, we have deleted CKD2 row from table 2, please find new table 2.
b. “Reading the text I could understand that 10 patients in the non-severe group and 41 in the severe one had no renal recovery”
We agree with his calculation about number of patients with renal recovery. This number is in accordance with our text and table 1. In table 1, 68/78 patients had renal recovery in the non-severe (78-10 =10 had no renal recovery) group and 22/63 (63-22 =41 had no renal recovery) in the severe group.
c. “moreover adding 34+46+16+9 in the non-severe group we obtain 105, but 105+41 is non 112.The same happens in the severe group.”
Along with the 41 patients with no renal recovery, the Reviewer has added patients died before day 90 (46),after day 90 (16) and patients lost to follow-up (9) (n=46+16+9 = 71). However, he also included in his calculation the 34 patients died in ICU (resulting in 71 + 34= 105 patients). However, these 34 patients were already accounted for because they died either before day 90, or after day 90. We therefore maintain our calculation n= (46+16+9)+41 = 112. To avoid this misunderstanding, we have deleted “in ICU” from died section in the figure 1.
d. “I strongly disagree with calculation of chronic kidney disease in the two groups: 112 had non-severe AKI, of whom 9 had chronic kidney disease at three years (8%); 120 had severe AKI, of whom 14 had chronic kidney disease at three years (11.6%). These results sound different from those reported by authors.”
We have been following our population for 3 years. Since they had no chronic kidney disease at the time of inclusion, the development of CKD is an incident event. 109/139 patients (78%) died before day 90 which is the time limit after which CKD can be defined. In the 30 remaining patients, we only considered this event at 3 years and could not determine whether CKD developed in the patients who will further die. We agree this could be disturbing if CKD has not been detected in the meantime, preventing its specific management and favoring death. We suppose these very severe patients were actually detected and treated accordingly. Our hypothesis is that less severe patients with a longer outcome need a specific follow-up to detect and to prevent CKD. This pragmatic view led us to identify incidence of CKD only in patients who survived at 3 years. To take into consideration your comment, we have specified that the result of the incidence of CKD was calculated, "amongst patients who survived” (line 230)
We also added in the discussion of the study limitations (line 334) “We could not determine whether CKD developed before death in the patients who died. However, 109/139 patients (78%) died before day 90 which is the time limit after which CKD can be defined. Our hypothesis is that less severe patients with a longer outcome need a specific follow-up to detect and to prevent CKD. This pragmatic view led us to identify incidence of CKD only in patients who survived at 3 years”.
ď‚· “In my opinion Figure 2 is related to patients’ survival, in fact the numbers reported at 90 days are the same of the deaths at 90 days, therefore the figure should be renamed from renal survival in patients’ survival.”
We fully agree with this comment, figure 2 was renamed.
ď‚· “The paragraph named “Renal specialist following” is useless”
One of the clinical impacts of this study could be to improve the long-term follow-up of patient who have suffered from non-severe AKI. It is written in these section that only half of CKD patients were followed by a kidney specialist suggesting that patients and probably the general practitioner were not aware they had CKD. We let the Editor decide whether this paragraph should remain or not in the discussion. But this is an outcome that was assessed by the investigator and which, despite it is a secondary outcome, carries an important message
ď‚· “References should follow the journal style.”
References was rewritten following the journal style.
ď‚· “Discussion should be rewritten taken into consideration risk factors for development of chronic kidney disease as suggested by different papers, eventually relating the items to critical ill patients.”
In line with another reviewer comments, the discussion was rewritten. This paragraph and 3 references suggested by reviewer 2 was added.
“Nevertheless, many factors may influence CKD development 3 years after an AKI in ICU and we cannot conclude whether non-severe AKI itself was independently implicated in the high CKD incidence at 3 years. First, AKI itself: in vitro studies have highlighted different mechanisms linking AKI and CKD such as persistent interstitial inflammation or tubular’s vascular damages. Secondly, individual factor can lead to the development of CKD at long-term. For example, diabetes is associated with a decrease in kidney function in many studies. In the multivariate analysis, diabetes was a risk factor for a decline in kidney function in AKI patients without prior CKD. These results were already demonstrated in non-critically ill patient. However, many other clinical conditions are associated with the risk of developing CKD such as age, sex, diabetes, hypertension, albuminuria, initial eGFR, high triglyceride levels and low HDL cholesterol levels[17][18]. In a future study, the Kidneyfailurerisk.com Canadian score which estimates the risk to develop CKD at 2 or 5 years and is well validated in a variety of populations [19] could be used to compare the expected incidence assessed by this score with the observed increased incidence of CKD.”
References added:
17. O'Seaghdha, C. M.; Lyass, A.; Massaro, J. M.; Meigs, J. B.; Coresh, J.; D'Agostino, R. B.; Astor, B. C.; Fox, C. S. A risk score for chronic kidney disease in the general population. Am. J. Med. 2012, 125, 270–277.
18. McMahon, G. M.; Preis, S. R.; Hwang, S.-J.; Fox, C. S. Mid-adulthood risk factor profiles for CKD. Journal of the American Society of Nephrology : JASN 2014, 25, 2633–2641.
19. Tangri, N.; Grams, M. E.; Levey, A. S.; Coresh, J.; Appel, L. J.; Astor, B. C.; Chodick, G.; Collins, A. J.; Djurdjev, O.; Elley, C. R.; Evans, M.; Garg, A. X.; Hallan, S. I.; Inker, L. A.; Ito, S.; Jee, S. H.; Kovesdy, C. P.; Kronenberg, F.; Heerspink, H. J. L.; Marks, A.; Nadkarni, G. N.; Navaneethan, S. D.; Nelson, R. G.; Titze, S.; Sarnak, M. J.; Stengel, B.; Woodward, M.; Iseki, K.; CKD Prognosis Consortium Multinational Assessment of Accuracy of Equations for Predicting Risk of Kidney Failure: A Meta-analysis. JAMA : the journal of the American Medical Association 2016, 315, 164–174.
And : kidneyfailurerisk.com

New Figure 1
New table 2
CKD stages
At 3 years
Non-severe AKI
at inclusion
n=41 (%)
Severe AKI
at inclusion
n=30 (%)
Total
n=71 (%)
CKD3 (60<eGFR<30)
7 (17)
10 (33)
17 (24)
CKD4 (30<eGFR<15)
2 (5)
1 (3)
3 (4)
CKD5 (eGFR<15)
0
3 (10)
3 (4)

Reviewer 3 Report

General Comment

The authors conducted a prospective observational study on the prognosis of non-severe AKI and found that patients with non-severe AKI frequently develop CKD. Although the current study is a small-scale observational study at a single center, it shows the necessity of long-term follow-up of patients after AKI, and this information would be clinically valuable. This paper is well written and the study design and analysis methods seem to be appropriate. However there are minor errors are found, the authors should revise them.

Specific Comments

The authors should correct the error. In the abstract “From 20013 to 2015” ⇒ “From 2013 to 2015”.

In Abstract, the authors state that “Patients with prior eGFR <90ml/min were excluded.” This information may mislead readers that the authors analyzed data using eGFR without body surface area correction. I think the unit of eGFR should be expressed as ml/min /1.73m2.

Author Response

Reply to the Review Report

Reviewer 3

We thank the reviewer 3 for his positive comment of our paper.

We have made the change he suggested

The authors should correct the error. In the abstract “From 20013 to 2015” ⇒“From 2013 to 2015”

We have corrected this mistake

In Abstract, the authors state that “Patients with prior eGFR <90ml/min were excluded.” This information may mislead readers that the authors analyzed data using eGFR without body surface area correction. I think the unit of eGFR should be expressed as ml/min /1.73m2

We fully agree with this comment, correction has been done.

Round 2

Reviewer 1 Report

Thanks for your response. I have no other comment.

Reviewer 2 Report

The authors have aknowledged all may concerns.